

# A fracture mechanics framework for optimising design and
# inspection of offshore Wind Turbine support structures against
# fatigue failure
Peyman Amirafshari[1], Feargal Brenan[1], Athanasios Kolios[1]
[1]Department of Naval Architecture, Ocean and Marine Engineering, University of Strathclyde,
Glasgow, G4 0LZ, United Kingdom
*Correspondence to*: Peyman Amirafshari (amirafshari.peyman@strath.ac.uk)

## 9 Abstract

Offshore Wind Turbine (OWT) support structures need to be designed against fatigue failure
under cyclic aerodynamic and wave loading. The fatigue failure can be accelerated in a corrosive
sea environment. Traditionally, a stress-life approach called the S-N curve method has been
used for design of structures against fatigue failure. There are a number of limitations in S-N
approach related to welded structures which can be addressed by the fracture mechanics
approach. In this paper the limitations of the S-N approach related to OWT support structure
are addressed, a fatigue design framework based on fracture mechanics is developed. The
application of the framework to a monopile OWT support structure is demonstrated and
optimisation of in-service inspection of the structure is studied. It was found that both the design
of the weld joint and Non-destructive testing techniques can be optimised to reduce In-service
frequency. Furthermore, probabilistic fracture mechanics as a form of risk-based design is
outlined and its application to the monopile support structure is studied. The probabilistic model
showed to possess a better capability to account for NDT reliability over a range of possible
crack sizes as well as providing a risk associated with the chosen inspection time which can be
used in inspection cost benefit analysis. There are a number of areas for future research.
including better estimate of fatigue stress with a time-history analysis, the application of
framework to other types of support structures such as Jackets and Tripods, and integration of
risk-based optimisation with a cost benefit analysis.

## 28 1 Introduction

Wind turbines are playing a key role in decarbonising world power production system. Target
share of energy from renewable sources in European Union (EU) countries set out by National
Energy and Climate Plans (NECPs) is aimed to reach 32% by 2030 and 100% by 2050. In 2018
the total share of energy from renewable sources were 18% in EU and 16% in United Kingdom
(European Environment Agency, 2019). Thanks to commitment of European countries to
achieve the above targets the prospects for the offshore renewable industry for further growth
continues to be strong (Fraile et al., 2019).
Since the power production of a wind turbine is directly related to the wind velocity at the hub,
the developments of Offshore Wind Turbine (OWT) are expected to grow in order to harvest
more power from offshore sites where wind speed is generally higher compared to the onshore.
Despite their higher wind power capacity, the biggest disadvantage of OWTs is their
construction and maintenance costs. Due to their remote location their inspection and
maintenance is challenging and expensive. Therefore, optimising design and maintenance of





these structure can decrease the levelized cost of electricity (LCOE) (Baum et al., 2018) and
(Luengo and Kolios, 2015).
OWT support structures constantly experience cyclic stress imposed by wind turbulences and
wave loading which makes them prone to the fatigue failure (Barltrop and Adams, 1991). The
fatigue damage accumulation could be further accelerated if exposed to the corrosive marine
environment.
There are two approaches for quantifying fatigue damage: The S-N (Stress vs. Number of cycles)
method and the Fracture Mechanics (FM) approach.
Standards such as IEC 61400-3 (IEC, 2009), DNVGL-ST-0126 (DNVGL, 2016a), DNVGL-ST-
0437 (DNVGL, 2016b) and DNVGL-RP-C203 (DNV, 2010) are commonly used for the design of
offshore wind turbines against fatigue failure. Current design approaches are solely based on
the S-N method. In this approach fatigue life of a structural element is determined using a
relevant S-N curve, recommended by one of the standards or derived from bespoke fatigue test
programs. Service induced stresses, contributing to fatigue damage accumulations, are
determined from structural analysis then a suitable joint class capable of resisting those
stresses is specified. Alternatively, if the joint class is known, maximum allowable fatigue
stresses for the intended life of the structure is determined from the relevant S-N curve
(Hobbacher, 2008).
Fatigue design of steel structures using S-N data is commonly preferred to the Fracture
Mechanics approach due to its simplicity (Naess, 1985). The S-N approach is also considered
more reliable since it is based on fatigue test compared to the Fracture Mechanics which is
based on calculations where additional input variables (e.g. crack growth rate, toughness, and
residual stress distributions) need to be considered (Anderson, 2005).
Despite its popularity, a number limitations exist with the S-N data approach in relation to
offshore wind turbine structures:
**Design for inspection:** Many structures are designed considering a damage tolerant philosophy
where the structure is expected to tolerate certain levels of fatigue damage until next scheduled
inspection (Fig. 1). The expected crack size at the time of the inspection is estimated using
Fracture Mechanics and a suitable non-destructive testing (NDT) technique capable of detecting
the critical crack size is prescribed. The S-N approach can only quantify the accumulated
damage without providing any information about the size and dimensions of the damage.
Fracture mechanics on the other hand estimates time-dependent fatigue crack size. In OWT
structures, due to access restrictions, the choice of NDT method can be limited to a certain NDT
method with a specific detection capability. Therefore, it may be necessary to consider the
Probability of Non-Detection (POND) and improve the design for such a scenario.



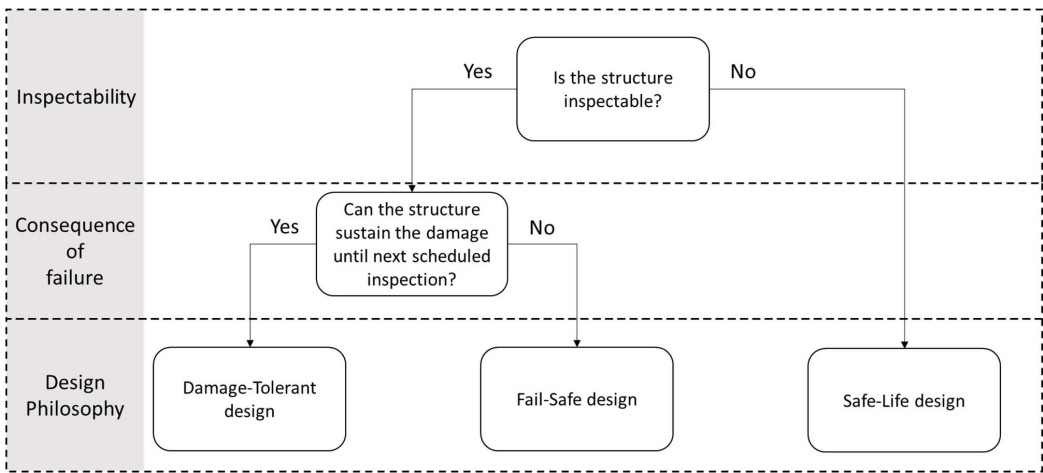

**Figure 1 Relationship between inspection and fatigue design philosophy**

**Effect of larger defect sizes:** S-N data is based on the assumption that the initial defect sizes are small, typically between 0.04 to 0.2 mm (BSI7608, 2015), assuming that an appropriate fabrication quality control program is in place which can detect larger fabrication defects. In practice, reliability and efficiency of such a program and the NDT techniques are uncertain and vary considerably among fabrication yards (Amirafshari, 2019). Assessment and design of the welded joints considering the presence of large defects is only possible using a Fracture Mechanics approach. An improved joint design can be achieved allowing for possible fabrication defects by, for example, specifying larger thicknesses, higher toughness steels, post weld heat treatment, etc (Zerbst et al., 2015).

**New welding processes:** There are always efforts to improve structural resistance, fabrication efficiency and weld quality by developing and implementing new welding technologies. Those processes may inevitably have altered characteristics (defect rates, sizes, and geometry, residual stresses, material toughness, etc.), which affect fatigue failure of the joint. Considering these variables using S-N data will require development of bespoke fatigue test program which is not always feasible (Lassen and Recho, 2013). A more efficient and cost-effective solution is the application of fracture mechanics.

**New materials:** development and use of new steel grades with higher tensile strength and weld consumable with superior weldability characteristics affects fatigue life. I.e. higher strength steel will be capable of resisting higher stresses, but the fatigue resistance does not increase proportionally (Okumoto et al., 2009). Contrary to the S-N method, these variables can be directly considered in the fatigue life prediction using Fracture Mechanics.

**Shakedown, and compressive residual stresses:** Fracture failure of welded joints is directly related to weld residual stresses. Tensile residual stress reduce fatigue life by reducing fracture capacity and moving the compressive part of cyclic stress to the tensile stress region. Part of these stresses can be relived under service or fabrication loads, which is commonly known as the "shake-down" effect (Li et al., 2007). In pile foundations, on the other hand, since the structure is driven to the soil a considerable amount of compressive residual stresses are induced into the pile (Da Costa et al., 2001), which can potentially improve the fatigue and fracture performance. The effect of compressive residual stress and the shakedown phenomena can be addressed using a fracture mechanics approach.

In this paper the fracture mechanics principals is briefly described, then a framework for an optimised design of structures based on fracture mechanics is developed. Then, probabilistic



fracture mechanics for risk and reliability-based design approaches is outlined. Finally,
application of the developed methods to a Monopile support structure is demonstrated.

## 2 Fracture Mechanics Approach

Fatigue cracks in welded structures initiate from weld fabrication defects at the joints. Even
sound welded joints often contain small undercuts (Fig. 2).
Fracture mechanics approach uses the Paris equation to predict crack growth under cyclic
stress. The method is based on the assumption that an initial flaw is present at the structure.
The initial flaw size depends on the rigour of the fabrication quality control (QC) program
(Jonsson et al., 2013). The reliability of the NDT method that is used during the QC, the extent
of the inspection (100% or partial) and the flaw acceptance criteria will influence such a rigour.
The fracture mechanics enables efficient application of NDT methods for in-service inspection
by specifying inspection interval(s) and the most effective NDT which has the capability of
reliable detection of the predicted crack size with a required confidence. This is illustrated in
Fig. 2 below, where the NDT inspection ($I_1$) detects cracks greater than initial flaw size ($a_0$). If
all such cracks are found and repaired the crack growth curve will be shifted down.

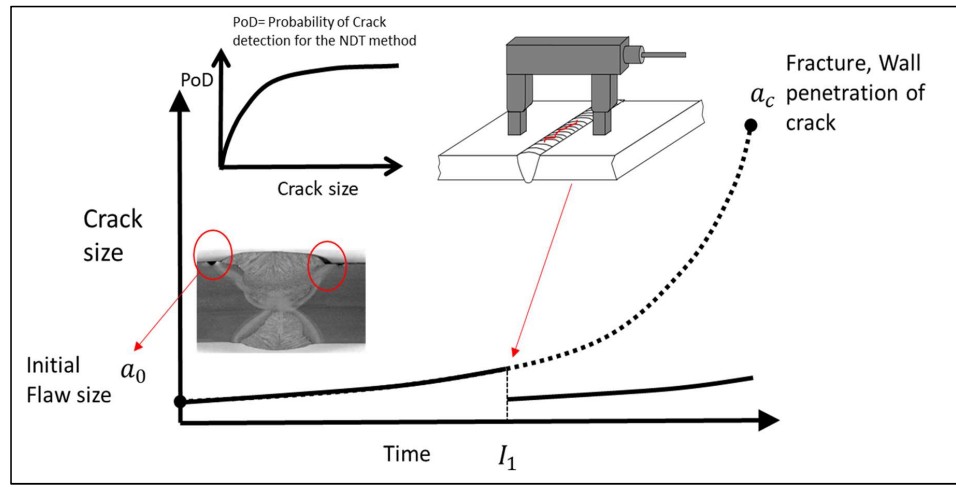

**Figure 2 Crack growth curve diagram**

### 2.1 Crack growth prediction

Fracture mechanics (FM) enables prediction of crack propagation by using the crack growth
rate, illustrated in Fig. 3. Region A is where crack growth rate occurs as soon as $\Delta K \geq \Delta K_{th}$ ,
where $\Delta K_{th}$ is the threshold value of $\Delta K$. The threshold value depends on a number of factors
such as the stress ratio $= K_{max}/K_{min}$ , sequence effect, residual stresses, loading frequency, and
the environment. Region B is where the crack growth rate increases with $\Delta K$ to a constant
power. Region C is where the crack growth rate increases rapidly until failure occurs as soon as
$K \geq K_{critical}.$



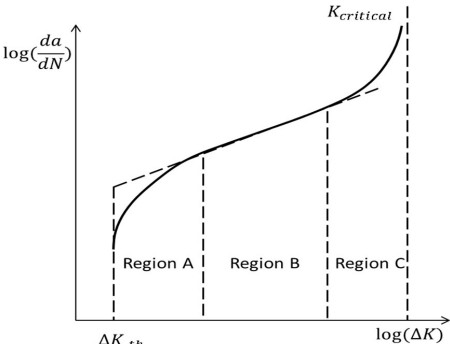


**Figure 3 Schematic of crack propagation curve according to Paris-Erdogan law** (Amirafshari, 2019)

In the FM approach crack growth rate is commonly described by the Paris-Erdogan Eq. (1):

$$\frac{da}{dN} = C * \Delta K^m \qquad (1)$$

where, $\frac{da}{dN}$ is the rate of crack growth with respect to load cycles, $\Delta K$ is the change in stress
intensity factor, and C and m are material constants. Recently a bilinear crack growth model
has been used, as well (Fig. 4). BS7910:2015 (BS7910, 2015a) recommended model is the
bilinear model, while the simplified model is cited, as well.

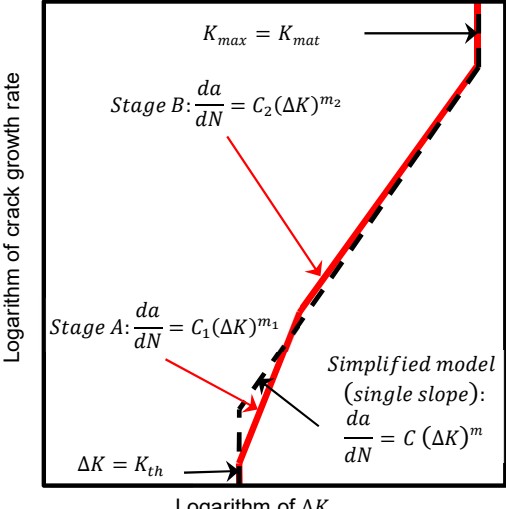

**Figure 4 Schematic of crack growth models by Paris law**
Stress intensity factor is described by:

$$\Delta K = Y\sigma\sqrt{\pi a} \qquad (2)$$

where, $a$ is flaw size, $\sigma$ is stress at the flaw, and $Y$ is the geometry function which depends on
both the geometry under consideration and the loading mode. There are several ways in which
solutions for $Y$ can be obtained. Although it is possible to derive solutions for simple geometries





analytically, e.g. using 'weight functions', numerical techniques are more commonly used (finite
elements, finite difference or boundary elements methods).
The number of cycles to failure is calculated by rearranging and nitrating Eq. (1):

$$N = \int_{a_0}^{a_f} \frac{da}{C(\Delta K)^m} = \frac{1}{A * Y^m * \Delta\sigma^m * \pi^{\frac{m}{2}}} * \frac{a_f^{\left(1-\frac{m}{2}\right)} - a_0^{\left(1-\frac{m}{2}\right)}}{1-\frac{m}{2}} \qquad (3)$$

Offshore structure are not subjected to constant amplitude stress, but a variable amplitude
stress spectrum. If the long-term stress distribution is converted into a step function of $n$ blocks
generally of equal length in log $N$, the crack size increment for the step i is:

$$\Delta a_i = C(\Delta K_i)^m \Delta N_i \qquad (5)$$

moreover, the final crack size at the end of the $N$ cycles is obtained by summing Eq. (5) for the
$n$ stress blocks:

$$a_N = a_0 + \sum_{i=1}^{N} \Delta a_i \qquad (6)$$

Equation (5) is only valid for small values of $\Delta a_i$ since $\Delta K_i$ depends on the crack size, which
requires dividing the stress range spectrum into a large number of stress blocks.
The number of cycles to failure may, alternatively, be calculated according to Eq. (7) using an
equivalent constant amplitude stress ranges $\Delta\sigma_{eq}$ giving the same amount of damage (Naess,
161 1985):

$$\Delta\sigma_{eq} = \left[\int_0^\infty \Delta\sigma^\beta \, p_{\Delta\sigma}(\Delta\sigma) d\Delta\sigma\right]^{1/\beta} \qquad (7)$$

where β is the contribution factor. For the central part of the crack growth curve β is often taken
as the slope of the of the crack growth line. $p_{\Delta\sigma}(\Delta\sigma)$ is the probability density function of stress
range $\Delta\sigma$.

## 2.2 Failure criteria

### 2.2.1 Through thickness

In the through-thickness criterion, the initial fatigue crack is assumed to be a surface breaking
flaw growing along the height ($a$) and length ($2C$) of the flaw. The failure happens when the
crack height penetrates through the thickness of the wall (Fig. 5). This criterion is, particularly,



commonly adopted for structures containing pressurised containments e.g. pipe lines, pressure
vessels, etc.

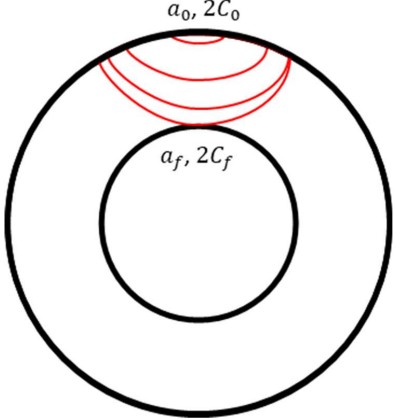


**Figure 5 Diagram of a surface crack penetrating wall**

### 2.2.2 Total Collapse criteria

Many structures have the capacity to sustain through-thickness cracks until the crack length
reaches a critical length. Thin wide plates that are primarily subjected to membrane stress and
redundant structures such as jacket type platforms, and stiffened plate hull structures are
examples of such structures.

In structural reliability analysis the probability of a collapse can be considered as a probability
of a fatigue crack failure, $P_F$, times the probability of a collapse given that there is a fatigue
failure in the structure, $P_{SYS}$ . The probability of the total structural collapse due to fatigue failure
should be below a target probability of failure, $P_t$:

$$P_F * P_{SYS} \leq P_t$$

(1)

For jacket structures the method of removing one member has been commonly used to assess
the residual capacity against overall collapse (DNV, 2015).

### 2.2.3 Critical crack size

The fatigue failure is considered to occur when the crack size reaches a critical value. There are
generally two ways to determine the critical size, which is explained in the coming sections:

1. Based on geometry of the structural member
2. Based on Failure Assessment diagram

The critical size maybe then reduced to account for further safety factors.

#### 2.2.3.1 *Based on geometry of the structural member*

For ductile structures, it is common to take the material thickness as the critical crack height
$(a_f = a_{cr} = Thickness)$. However, normally the assumption is that the crack grows under cyclic
loading which corresponds to normal service loading until it becomes through thickness. In
reality, failure often happens during extreme load occurrences. The cracked structure may fail





under such extreme loading through failure of the thickness ligament (Fig. 6). The brittle or
elasto-plastic ligament failure may also occur in structures with low fracture toughness.

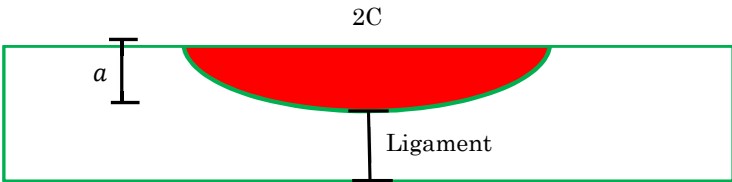


**Figure 6 Diagram of the remaining ligament in a semi-spherical crack**
To address above limitation the failure assessment diagram (FAD) may be adopted.
*2.2.3.2   Based on Failure Assessment Diagram (FAD)*
Failure Assessment Diagram (FAD) can assess the failure of the through-thickness crack as
well as implementing extreme load occurrences by treating them as the primary stress. The
approach is explained below.
When a crack propagates through a structure, ultimately the crack size reaches a critical size
$a_f$. $a_f$ corresponds to a critical stress intensity factor, usually taken as characteristic of the
fracture toughness $K_{mat}$, at which fracture happens. Alternatively, if the applied load is high
and structure tensile strength is low, the structure may reach its tensile strength capacity and
fail by plastic collapse. The latter is more favourable as it is usually associated with large
deformations prior to failure providing some level of warning. In between brittle fracture and
global collapse is an elastoplastic failure mode, where failure occurs before reaching the plastic
capacity or toughness limit; this has been best described by failure assessment diagram (FAD)
in the R6 procedure in 1976 and improved over time by e.g. including the options available to
model specific material properties. The body of knowledge encapsulated in R6 affected the
development of British Standards documents in various ways over the years, leading to
BS7910:1999 and the latest version at the time of writing, (BS7910, 2015a).
The failure assessment line (FAL) represents the normalised crack driving force:

$$K_r = \frac{K_{elastic}}{K_{elastic\ plastic}}$$    (8)

$K_r$ is equal to 1 where applied load is zero and declines as the ratio between applied load and
yield load ($L_r$) increases towards collapse load (see Fig. 6).
The plastic collapse load is calculated based on yield stress. However, the material has further
load carrying capacity as it work-hardens through yield to the ultimate tensile stress. To take
this into account the rightwards limit of the curve is fixed at the ratio of the flow stress to the
yield stress:

$$L_r = \frac{\sigma_{flow}}{\sigma_Y}$$    (9)

The flow stress is the average of the yield and ultimate stresses:



$$\sigma_{flow} = \frac{\sigma_Y + \sigma_U}{2} \tag{10}$$

If the assessment point lies inside the envelope (below the FAL), the fracture mechanics driving
parameter is lower than the materials resistance parameter and the part should be safe,
otherwise there is a risk of failure. The failure assessment diagram can be determined with one
of the procedures provided by (BS7910, 2015a). As it is illustrated in Fig. 6, FAD may be
categorised into three different zones: Zone 1 is the fracture dominant zone, Zone 2 is the
elastoplastic region or the knee region, and Zone three is the collapse dominant zone.
(BS7910, 2015a) has three alternative approaches Option 1, Option 2 and Option 3. These are
of increasing complexity in terms of the required material and stress analysis data but provide
results of increasing accuracy.
Option 1 (BS7910, 2015a) is a conservative procedure that is relatively simple to employ and
does not require detailed stress/strain data for the materials being analysed. The Failure
Assessment Line (FAL) for the Option 1 analysis is given by:

$$K_r = f(L_r) = (1 + 0.5 * L_r^2)^{-0.5} * (0.3 + 0.7 * \exp(-\mu * L_r^6)) \tag{11}$$

for $L_r < 1$, where: $\mu = min\left[0.001\frac{E}{\sigma_Y}; 0.6\right]$.
and:

$$K_r = f(L_r) = f(1)L_r^{(N-1)/2N} \tag{12}$$

For,$1 < L_r < L_{r,max}$, where N is the estimate of strain hardening exponent given by: $N = 0.3(1 -$
$\frac{\sigma_Y}{\sigma_{UTS}}$). and $L_{r,max} = \frac{\sigma_{flow}}{\sigma_Y}$.
Option 2A/3A of BS 7910:2005 generalised FAD, is similar but not identical to Option 1 (BS7910,
2015a)

$$K_r = (1 - 0.14 * L_r^2) * (0.3 + 0.7 * \exp(-0.65 * L_r^6)) \tag{13}$$

The BS7910:2015 Option 2 FAD is based on the use of a material-specific stress-strain curve.
The assessment line can be written as:

$$K_r = f(L_r) = \left[\frac{E\varepsilon_{ref}}{L_r\sigma_Y}, \frac{L_r^3\sigma_Y}{2E\varepsilon_{ref}}\right]^{-0.5} \tag{14}$$

$\varepsilon_{ref}$ is the true strain obtained from the uniaxial tensile stress-strain curve at a true stress
$L_r\sigma_Y$.
The option 3 failure assessment curve is specific to a particular material, geometry and loading
type using both elastic and elastic-plastic analyses of the flawed structure It is given by:

$$f(L_r) = \sqrt{\frac{J_e}{J}}, \text{ for } L_r < L_{max} \tag{15}$$




$$f(L_r) = 0, \text{ for } L_r > L_{max} \tag{16}$$

$J_e$ is the value from the J-integral from the elastic analysis at the load corresponding to the
value $L_r$. The Option 3 curve is not suitable for general use. It is useful only for specific cases as
an alternative approach to Options 1 and 2 (BS7910, 2015a).
Options 1&2 (BS7910, 2015a) and Option 2A/3A (BS7910, 2015a) for structural steel with
tensile stress of 550 MPa and Yield stress of 450 MPa are illustrated in Fig. 6. It can be seen
that the greatest difference between the three plotted locus is in the collapse region.

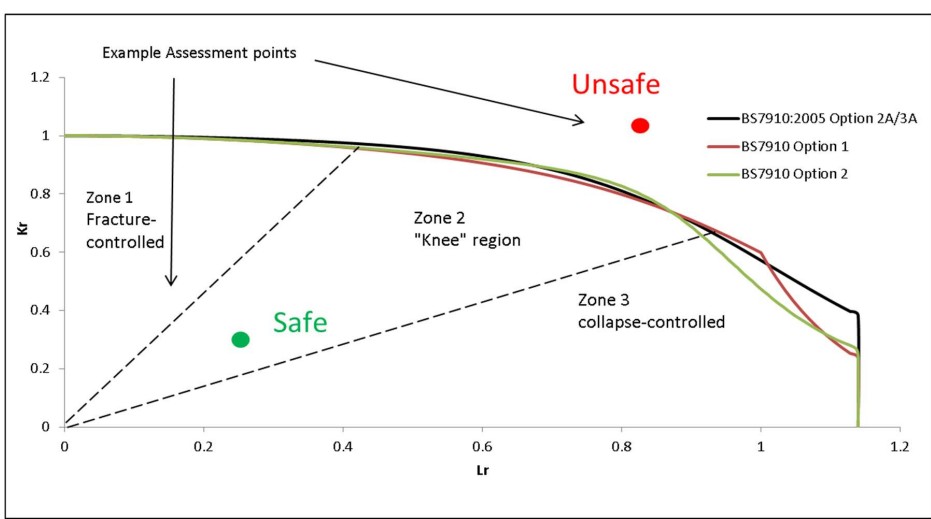

**Figure 7 Failure Assessment Diagram (FAD)** (Amirafshari, 2019)

## 3   Fracture Mechanics framework for structural design

The common practice in structural design is to specify dimensions of the structural component
based on the most critical limit state, usually ultimate limit state (ULS), and check or modify
the design based on other limit states such as serviceability limit sate (SLS) or fatigue limit
state (FLS).
In OWT support structures fatigue failure initiates from the welded connection, thus, the
fatigue design often involves prescribing local improvements to the welded connection. However,
since fatigue life is related to dynamic characteristics of the structure the global dimensions of
the structure may also need alterations to achieve higher fatigue resistance.
The fatigue damage prediction model could be the S-N curve method or the Linear Elastic
Fracture Mechanics (LEFM). Here, a LEFM method is adopted to address the limitations of the
S-N curve method. Fig. 7  shows the proposed framework.
First, the required inputs, such as structural dimensions (determined by structural design
based on ULS), initial flaw size, material toughness and tensile properties, stress at the flaw,
and parameters of Paris equation, are determined, the using the Paris equation for a chosen
increment of time ($N_i$), the increase in initial crack size is estimated. The predicted crack size is
then compared against failure criteria. The procedure is repeated for the next time increment
until the failure. If the failure is predicted to occur before intended life of the structure the



fatigue life may be enhanced by changing variables that affect the fatigue failure such as
structural dimensions, quality control requirements (initial flaw size), post fabrication
improvements (e.g. post weld heat treatment ), or by specifying inspection interval(s).

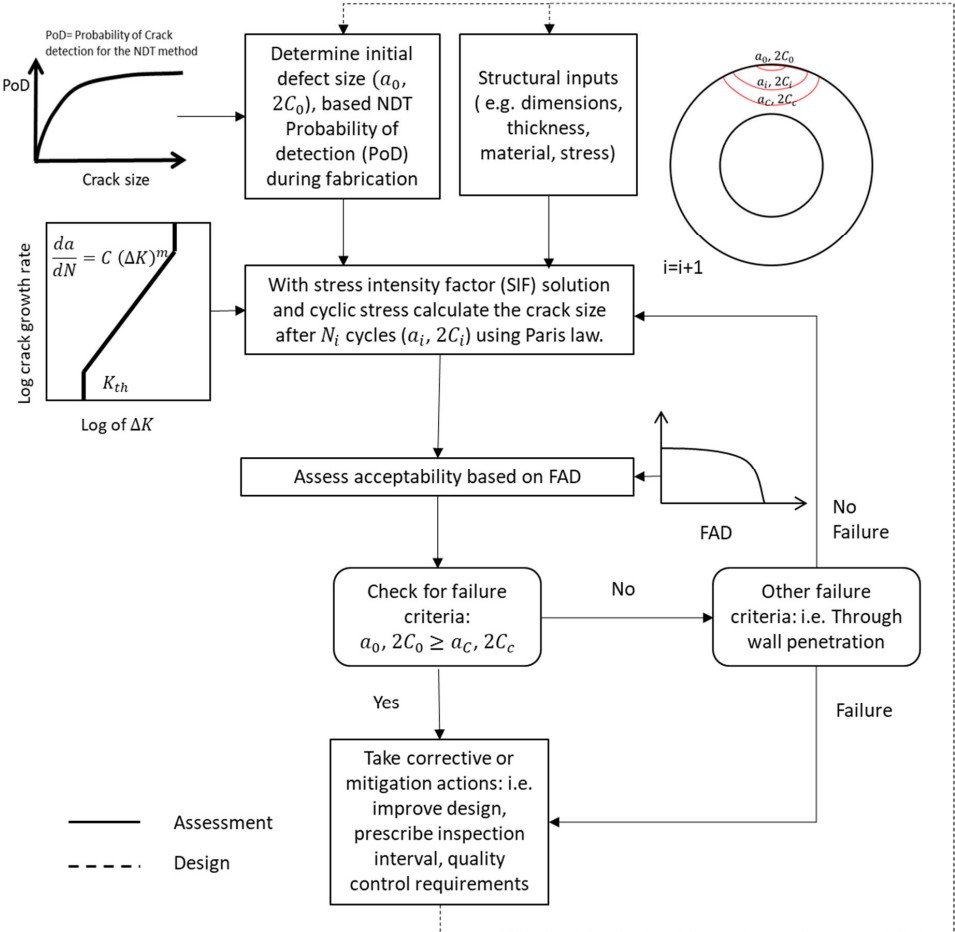

**Figure 8 Fracture Mechanics flow diagram for assessment and design of structures against fatigue failure**
### 3.1  Damage-tolerant design
The term damage-tolerance fracture mechanics normally refers to a design methodology in
which fracture mechanics analyses predict remaining life, and specifies inspection intervals.
This approach is typically applied to structures prone to time dependent crack growth. The
damage tolerance philosophy allows flaws to remain in the structure, provided they are well
below the critical size.
Once the critical crack size has been estimated, a safety factor is applied to determine the
tolerable flaw size $a_t$. The safety factor should be based on uncertainties in the input parameters
(e.g. stress, parameters in the Paris equation and toughness). Another consideration in
specifying the tolerable flaw size is the crack growth rate; $a_t$ should be chosen such that da/dt
at this flaw size is relatively small, and a reasonable length of time is required to grow the flaw
from $a_t$ to $a_c$ (Anderson, 2005). This is shown schematically in Fig. 8.



WIND ENERGY SCIENCE DISCUSSIONS

european academy of wind energy

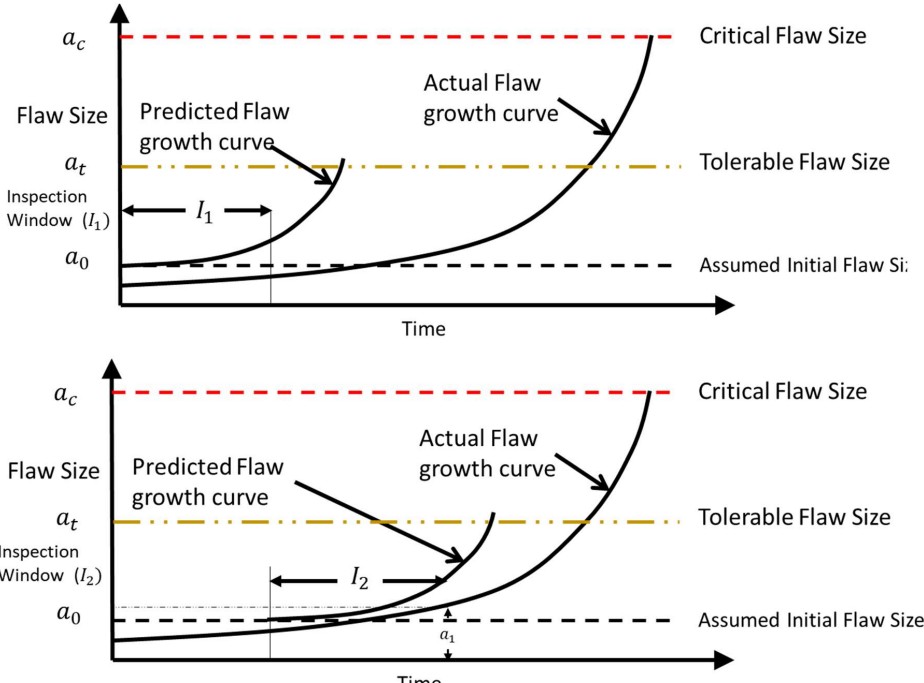


**Figure 9 schematic representation of damage tolerant fracture mechanics approach, adapted from** (Anderson, 2005)

### 3.2 Inspection reliability (PODs)

NDT techniques can only detect a limited number of defects of a certain size. For instance, an NDT method with 50% probability of detection at a certain size, is expected to miss 50% of the defects of that size, in other words, the real number of the defects with that size is likely to be 100% more than the detected. In structural integrity assessment, it is often convenient to plot detection probability against defect size, which constructs the so-called probability of detection curve (Fig. 10). Detection capabilities of NDT methods are directly related to the sizing of flaws (Georgiou, 2006). The bigger the flaw sizes, the more likely that they are detected. Fig. 9 shows the relationship between detected defect size distribution, the probability of detection of defect sizes and the actual defect size distribution that are present in the structure.

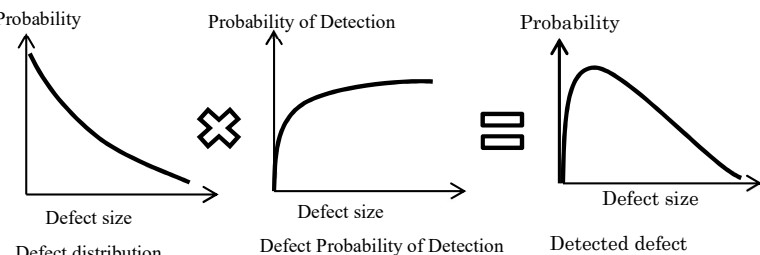

305

**Figure 10 Relationship between crack size distribution, Probability of detection and detected crack size distribution** (Amirafshari, 2019)

PoDs for NDT methods are highly dependent on various factors such as, the operator skills, testing environment, test specimen (thickness, geometry, material, etc.), type of the flaw, orientation and location of the flaw (Førli, 1999). Hence, accurate estimation of PoD curves



requires individual PoD test programs for specific projects. However, a number of lower bound
generic models are available in the literature for some specific NDT methods. Two of such
models, that are relevant to this work, are given in Fig. 10 and Table 1 below.
Further information about derivation, application and limitations of PoD can found in
(Georgiou, 2006).

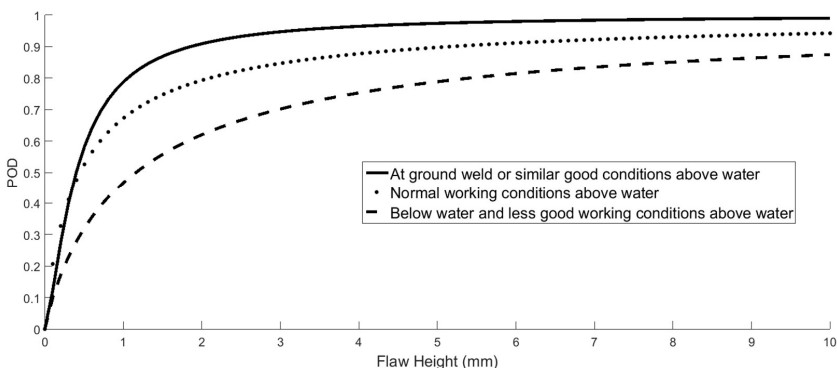

**Figure 11 DNV POD for surface NDE. Replotted from** (DNV, 2015)

| Method | Condition | | Flaw Length mm | Flaw through-thickness mm |
|---|---|---|---|---|
| Magnetic Particle Inspection (MPI) | Machined or ground | | 5 | 1.5 |
| | As-welded | With local dressing | 10 | 2 |
| | | With poor profile | 20 | 4 |
| Ultrasonic Testing (UT) | Convectional | | 15 | 3 |

**Table 1 NDT Reliability** (BS7910, 2015b)
## 3.3  Inspection strategy
Fracture mechanics assessment is closely tied to inspection method. The inspection method
provides input to the fracture mechanics assessment, which in turn helps to define inspection
intervals. A structure is inspected during construction for quality control purposes. Choice of
the NDT method varies between fabrication yards, but as a general rule all weldments are
visually inspected and may be complemented by inspection of limited number of checkpoints
using more reliable NDT techniques on a sampling basis (Amirafshari et al., 2018). If no
significant flaws are detected, the initial flaw size is set at an assumed value $a_0$, which
corresponds to the largest flaw that might be missed by NDE.
Generally, there are two strategies in inspection of structures that are susceptible to damage
mechanisms:
### 3.3.1  The inspection schedules are fixed (Periodic Maintenance):
Here, the fracture mechanics can be used to design the structure so that the possible fatigue
cracks remain below tolerable limits. The crack size at the time of inspection is predicted using
the Paris law in order to select an appropriate NDT method.
### 3.3.2  Inspection schedule is not fixed (Condition Based Maintenance):
In this case, the inspection interval and the NDT method can be optimised in such a way that
the inspection results in a safer condition or a minimised cost of maintenance and failure.



### 3.4 Design inputs

Design inputs can be categorised into design constraint(Table 2) and design variables (Table 3).
Here, only design variables related to a fracture mechanics method are considered. Further
information about design of offshore wind turbine support structures can be found in (Arany et
al., 2017) and (Van Wingerde et al., 2006).
Depending on chosen maintenance strategy the inspection capabilities may be considered as
design constraint or design variable.
If a probabilistic approach is employed instead of the conventional deterministic approach, the
variables are considered stochastically and target probabilities of failures are used instead of
allowable deterministic values (Table 2).

| Design Constraint | | |
|---|---|---|
| Limit State | Deterministic | Allowable damage, stress, etc. |
| | Probabilistic | Target levels of reliability |
| Inspection capabilities | During fabrication | • Extend of inspection<br>• NDT PoD |
| | During service | • Inspection schedule (fixed periodic inspections)<br>• NDT method (e.g. POD, access restrictions, costs) |

**Table 2 Design constraints for damage tolerant fracture mechanics design**

| Design variables | Inspection and Monitoring options ( Condition Based Maintenance) | NDT methods |
|---|---|---|
| | | Condition monitoring |
| | Design options | Structural design options:<br>• Thickness<br>• Redundancy<br>• Material selection |
| | | Fabrication specifications:<br>• Weld profile improvements<br>• Post Weld Heat Treatment<br>• Quality Control(i.e. NDT during fabrication, Tolerance limits ) |

**Table 3 Design variables for damage tolerant fracture mechanics design**

## 4 Probabilistic Fracture Mechanics

Fracture mechanics approaches are commonly used deterministically and generally have a
hierarchical nature, i.e. the analyst may progressively reduce the level of conservatism in
assumptions by increasing the complexity level of the analysis and consequently the precision
of results until the operation of the structure is found to be fit-for-service. Otherwise, the
structure will require a repair, a reduction of service (for example lowering primary stress) or
resistance improvements (i.e. reduction of secondary stresses by stress relief techniques). This
type of approach is particularly useful in the assessment of safety cases where the aim is to
demonstrate that the structure is safe.
In deterministic analyses, uncertainty in variables are dealt with by taking upper bound and
lower bound of those variables- upper bound values of applied variables such as stress and flaw
size, with lower bound values of resistance variables such as fracture toughness. In reality, the
probability of all unfavourable conditions occurring at the same time is very low and often too

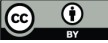



conservative. An alternative approach is a probabilistic analysis, in which, uncertain variables
are treated stochastically and as random variables.
In probabilistic assessments, all possible combinations of input variables leading to failure are
compared against total possible combinations, and a probability of failure is estimated instead
of a definite fail or not-fail evaluation. Probabilistic analysis is also in-line with the damage
tolerant philosophy. The failure probability for the limit state function may be estimated using
one of available analytical, numerical or simulation methods such Monte Carlo simulation.
Figure 12 shows Probabilistic fracture assessment using Monte Carlo method and based on the
FAD.

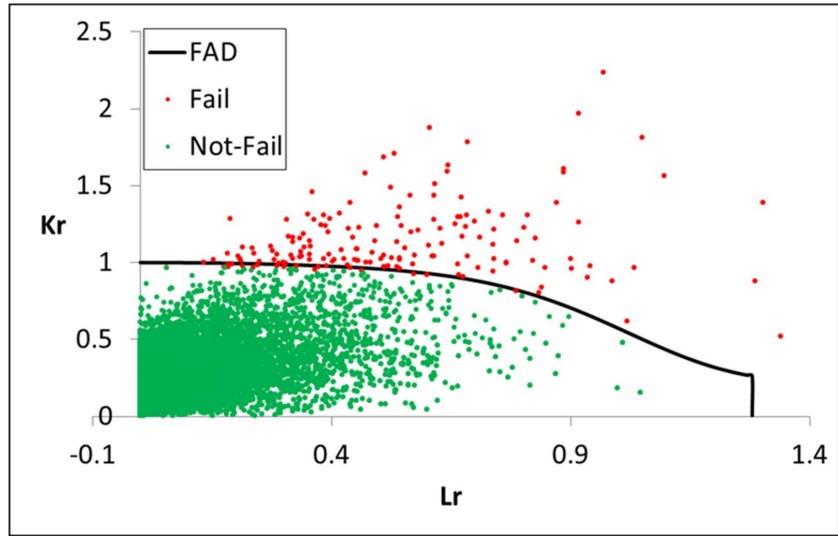

**Figure 12 Probabilistic fracture assessment using Monte Carlo method and based on FAD** (Amirafshari, 2019)
One limitation of deterministic fracture mechanics is that conservative prediction of critical
defect size and the time to the failure may reduce inspection efficiency by targeting wrong defect
sizes and at a wrong time in service, whereas probabilistic assessment will provide a more
efficient result (Lotsberg et al., 2016). Probabilistic failure assessment of the structures is also
known as Reliability analysis. These two terminologies are often used interchangeably.

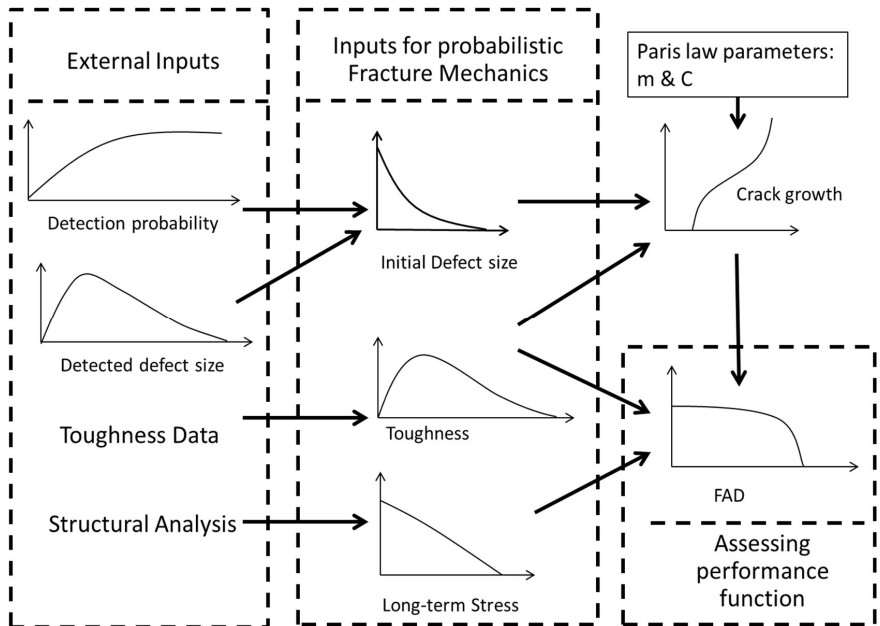

**Figure 13 A schematic presentation of the inputs to Probabilistic Fracture Mechanics** (Amirafshari, 2019)
Figure 18 shows schematic presentation of the inputs to probabilistic fracture mechanics.
Probabilistic fatigue and fracture analysis will predict the time-dependent failure probability of
the structure (Fig. 19). The predicted reliability will then need to be compared against an
appropriate target reliability level.

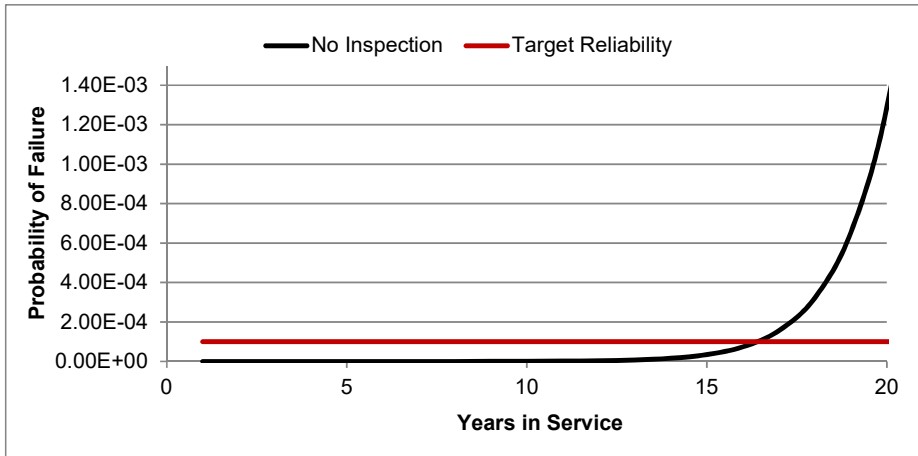

**Figure 14 Example of a time-dependent fatigue and fracture reliability curve**
## 4.1 Target reliability levels
Target reliability values may be employed to ensure that a required level of safety is achieved.
The target reliability measures depend on the failure consequence as well as the cost and effort
to reduce the risk of failure. The consequence of failure can be the risk of human injury and
fatality, economic consequence, and social impacts. The target reliability should always
correspond to a reference period, e.g. annual or service life probability of failure. If the relevant



consequence is the risk of human life, annual failure probabilities are preferred to ensure a
consistent level of tolerable risks at any time. Target reliabilities maybe defined in four different
ways:
1. The standard developers recommend a reasonable value. This method is used for novel
structures.
2. Reliability implied by standards. The level of risk is estimated for a design standard that is
considered to be satisfactory. This method has been commonly used for standard revisions,
particularly where the intention has been to provide a more uniform safety level for different
structural types and loading types. By carrying out a reliability analysis of the structure
satisfying a specific code using a given probabilistic model, the implicit required level in this
code will be obtained, which may be applied as the target reliability level. The advantage
with this approach compared to applying a predefined reliability level is that the same
probabilistic approach is applied in the definition of the inherent reliability of the code
specified structure and the considered structure, reducing the influence of the applied
uncertainty modelling in the determination of the target reliability level.
3. The target level for risk assessment based on failure experiences. This method is particularly
useful when the functional reliability of the system is more important than the reliability of
individual components. In the automotive industry or electronic components manufacturing
component reliability is determined by failure rate data of real components. The failure rate
data is then used in system reliability calculation(Bertsche, 2008).
4. Economic value analysis (cost-benefit analysis). Target reliabilities are chosen to minimise
total expected costs over the service life of the structure. In theory, this would be the
preferred method, but it is often impractical because of the data requirements for the model.
Examples of target reliabilities prescribed by codes and standards are listed in Table 6. For
further information about available models for developing target reliability levels for novel
structures reference is made to (Bhattacharya et al., 2001).

| | Scope | Limit state function | Minimum Reliability index | Maximum Probability of failure |
|---|---|---|---|---|
| Euro code. Basis of structural design (BSI, 2005) | buildings and civil engineering works | Ultimate limit states (ULS) | 3.3 to 4.3 for 50 years reference period and 4.2 to 5.2 for annual | $4.83 \times 10^{-4}$ to $8.54 \times 10^{-6}$ for 50 years reference period and $1.33 \times 10^{-5}$ to $9.96 \times 10^{-8}$ for annual |
| | Residential and office buildings, public buildings where consequences of failure are medium (e.g. an office building) | Fatigue limit state (FLS) | 1.5 to 3.8 for 50 years reference period | $6.68 \times 10^{-2}$ to $7.23 \times 10^{-5}$ for 50 years reference period |
| DNV (DNV, 1992) | Marine structures | | 3.09 to 4.75 | $1.00 \times 10^{-3}$ to $1.02 \times 10^{-6}$ |



| IEC61400-1 | Offshore Wind Turbines | ULS & FLS | 3.3 | $5.00 \times 10^{-4}$ |
|---|---|---|---|---|
| DNV_OS_J101 | Offshore Wind Turbines (unmanned structures) | ULS | | $1.00 \times 10^{-4}$ |
| DNV_OS_J101 | Offshore Wind Turbines (manned structures) | ULS | | $1.00 \times 10^{-5}$ |

**Table 7 Examples of target levels of reliabilities specified by standards**
## 4.2 Risk Based design
The purpose of risk analysis is to comprehend the nature of risk and its characteristics
including, where appropriate, the level of risk. Risk analysis involves a detailed consideration
of uncertainties, risk sources, consequences, likelihood, events, scenarios, controls and their
effectiveness. An event can have multiple causes and consequences and can affect multiple
objectives (ISO-31000, 2018). Risk remaining after protective measures are taken is called
residual risk (ISO-14971, 2012). The purpose of risk evaluation is to support decisions. Risk
evaluation involves comparing the results of the risk analysis with the established risk criteria
to determine where additional action is required (ISO-31000, 2018). The overall procedure for
risk analysis and risk evaluation is a risk assessment (ISO-31000, 2018).
A commonly used method of risk evaluation is the so-called Risk Matrix model in which the
failure probability is shown in one axis and the consequence of failure on the on the other. The
failure probability and consequence failure maybe specified quantitatively, qualitatively, or
semi-quantitatively, depending on the complexity of the model and the availability of data. Each
combination of failure probability and consequence of failure will then be assigned a
corresponding risk level. It is useful to show these levels in specific colour coding convention.
One such convention is an adapted traffic light convention in which low-risk levels are shown
in green, extreme risks in red and medium risk levels are coloured in yellow. It is also possible
to refine this colour coding further, for example, light yellow and dark yellow, to allow for more
risk levels. An example Risk Matrix is shown in Fig. 22.

| Probability of failure | 5. Frequent | HIGH | HIGH | EXTREME | EXTREME | EXTREME |
|---|---|---|---|---|---|---|
| | 4. Likely | MEDIUM | HIGH | HIGH | EXTREME | EXTREME |
| | 3. Possible | MEDIUM | MEDIUM | HIGH | HIGH | EXTREME |
| | 2. Unlikely | LOW | MEDIUM | MEDIUM | HIGH | HIGH |
| | 1. Rare | LOW | LOW | MEDIUM | HIGH | HIGH |
| | | 1. Negligible | 2. Minor | 3. Moderate | 4. Major | 5. Catastrophic |
| | | Consequence of failure | | | | |

**Figure 15 A typical Risk matrix diagram**
In order to assign an appropriate risk level (i.e. colour in the risk matrix) it is necessary to
establish risk acceptance levels. If a system has a risk value above the accepted levels, actions
should be taken to improve the safety through risk reduction measures. One challenge in this
practice is defining acceptable safety levels for activities, industries, structures, etc. Since the
acceptance of risk depends upon society perceptions, the acceptance criteria do not depend on
the risk value alone (Ayyub et al., 2002).





Another common risk evaluation method is the ALARP, which stands for "as low as reasonably
practicable", or ALARA (as low as reasonably achievable) (HSE, 2001). The ALARP basis is that
tolerable residual risk is reduced as far as reasonably practicable. For a risk to be ALARP, the
cost in reducing the risk further would be grossly disproportionate to the benefit gained. The
basis of ALARP is illustrated by the so-called carrot diagram in Fig. 23.

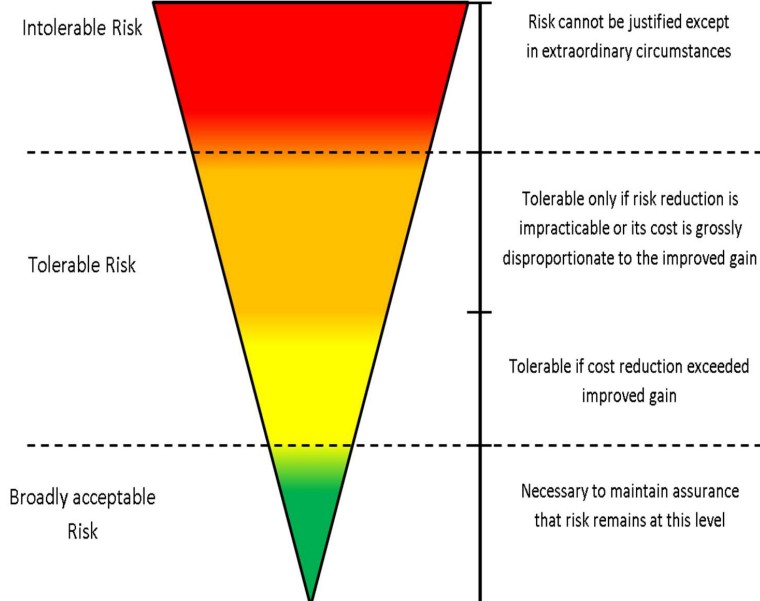

**Figure 16 ALARP Carrot diagram based on** (HSE, 2001)
By adopting a risk based approach in fracture mechanics for a chosen design parameter the
structural design may be assessed against the corresponding risk. As an example, the design
stress levels for a particular initial crack size will be associated with the corresponding risk
levels, as schematised in Fig. 24.

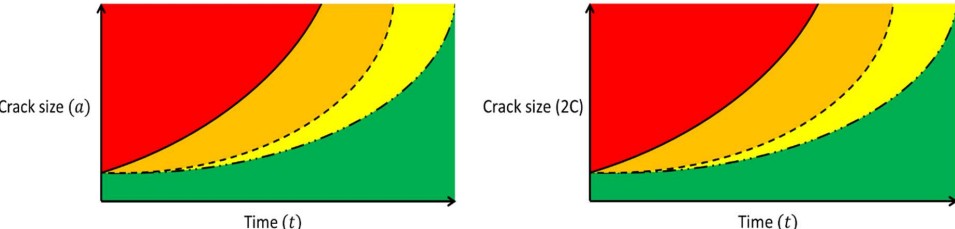

**Figure 17 schematics of Crack growth curves based risk profile**

## 5 Case-Study 1: Monopile OWT support structure

Fatigue design based of a baseline NREL 5MW offshore wind turbine (OWT) supported on a
monopile structure (Fig. 12) is presented here. The framework illustrated in Fig. 7 is used to
conduct the fracture mechanics assessment. Table 5 summarises inputs parameters used in this
study. Further information about the structure and the Finite Element Analysis can be found
in (Gentils et al., 2017).



Transverse butt weld (weld line perpendicular to the normal stress) are more prone to fatigue
damage than the longitudinal butt joints (weld line parallel to the normal stress). Figure 9
shows these joints in a monopole structure. A fatigue crack growing at the transverse butt weld
toe located in mud-line (Fig. 12) is considered as the most critical location.

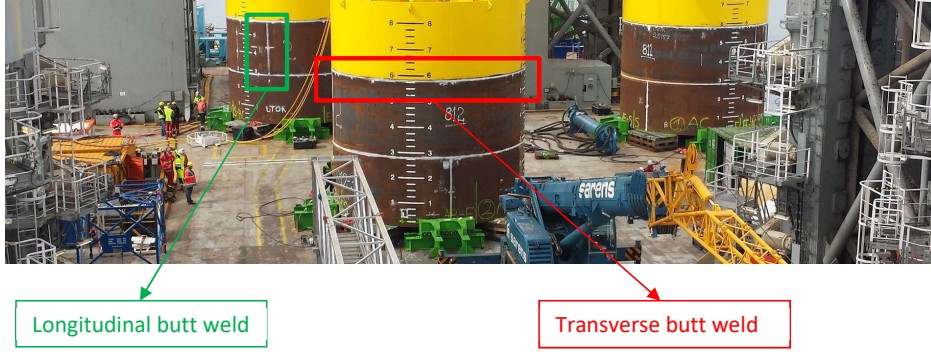

**Figure 18 Monopile welded connections (twd, 2019)**

| Case Description | | |
|---|---|---|
| Structure | NREL 5MW OWT | |
| Material Properties | Young Modulus | 210 |
| | Poisson Ratio | 0.38 |
| | Yield stress | 355 |
| | Tensile strength | 550 |
| | Toughness | 200 MPa* m^0.5 assumed |
| Fatigue assumptions | Crack growth model | Single slope Crack growth |
| | Cyclic stress | Equivalent constant amplitude stress 51.2 MPa |
| | Stress Intensity Solution | Surface flaw in a Plate |
| | Paris Law Constants | $m = 3.9, C = 3.814 * 10^{-16}$ for Crack growing in HAZ and in Air, $m = 3.3, C = 4.387 * 10^{-14}$ for Crack in HAZ and in with free corrosion, (for $da/dN$ in $mm/cycle$, and $\Delta K$, in $N/mm^{0.5}$), (Mehmanparast et al., 2017) |
| | Design cycles in life | $N_{life} = \eta_a * \eta_{rated} * (20\ [year] * 365[day\ per\ year] * [hour\ per\ year] * 60\ [min\ per\ hour])$, for this structure $= 1.253 * 10^8$ (Gentils et al., 2017) |
| Fracture assumptions | FAD | BS 7910 Option 1 |
| | Primary stress | 209 MPa |
| | Secondary stress | Weld Residual stress= 100 MPa, assumed |
| | Thickness (B) | 60 (mm) |
| | Initial Flaw dimensions (a*2C) | (1.5 mm * 5 mm) |

**Table 5 Inputs for Fatigue and fracture mechanics assessment**

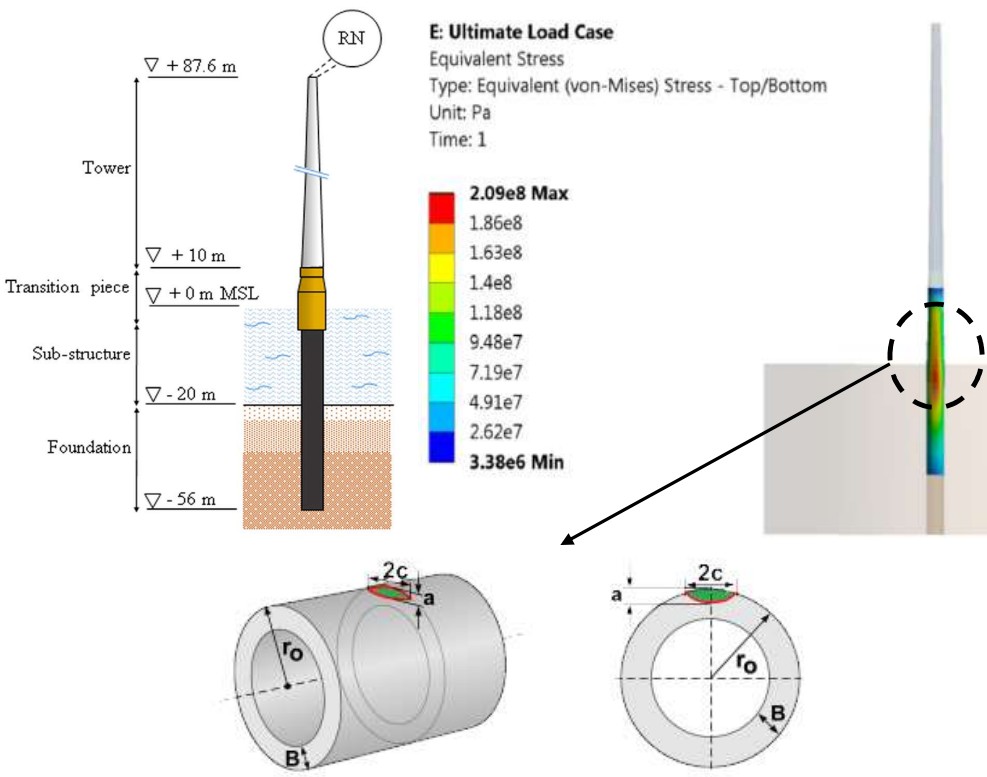


**Figure 19 The case study structure diagrams and FEA contour plots for the support structure**

Fatigue cracks normally initiate from small toe undercut weld defects (Fig. 2), thus, in this
study a semi-spherical flaw growing in heat affected zone (HAZ) of the joint is considered. NDT
inspection techniques are used during fabrication as part of quality control scheme. MPI and
UT are effective, and commonly used method to detect surface breaking and embedded flaws,
respectively. Here, initial flaw size is conservatively assumed to be equal to 90 % PoD the NDT
methods (Table 1). Primary fracture stress is taken as caused by ultimate limit state (ULS)
design stress (Fig. 12) corresponding to the parked wind turbine, under the 50-years Extreme
Wind Model (EWM) with the 50-years Reduced Wave Height (RWH) and Extreme Current
Model (ECM), defined as the Design Load Case (DLC) 6.1b and 2.1 for (IEC, 2019) and (DNV,
2013) standards, respectively. The crack growth stress is taken as the fatigue load case
corresponds to an operating state under Normal Turbulence Model (NTM) and Normal Sea
State (NSS) where wave height and cross zero periods are obtained from the joint probability
function of the site, assuming no current; it corresponds to the DLC 1.2 from the IEC standard
(IEC, 2019) and is assumed to represent the entire fatigue state (Gentils et al., 2017). Paris law
parameters reported by (Mehmanparast et al., 2017) for offshore wind monopile weldments has
been adopted. Other key assumptions and inputs for fatigue and fracture mechanics assessment
are given in Table 5.

## 5.1   Crack growth in Air

Crack growth parameters in Paris equation for ferritic steels depend on the, cyclic stress ratio,
and environmental condition (Amirafshari and Stacey, 2019). In presence of effective corrosion
protection measures, in-air conditions apply (BS7910, 2015a).



Fatigue and fracture assessment results for cracks propagation in air environment are given in Table 5. In a tolerant design, the tolerable crack sizes need to be selected way below critical sizes by considering some level of safety factors (Anderson, 2005). As described earlier, the chosen tolerable crack size needs to be determined in a region of crack size where crack growth rate with respect to time is small to allow for a long time before failure but large enough to be detected by the in-service inspection technique. Here, tolerable crack height of 5.2 mm is chosen which, depending on the inspection condition (Fig. 10), gives 70 to 90 percent Probability of Detection (PoD). As shown in Fig. 20, this will provide a good margin of safety and at least 6 years before failure (Fig. 22).

| Assessment results | | |
|---|---|---|
| Critical Crack size | $a_c = 45\ mm$ | $2C_c = 116\ mm$ |
| Tolerable crack size (Assumed) | $a_t = 5.2\ mm$ | $2C_t = 12\ mm$ |
| | $Lr_t = 0.592$ | $Kr_t = 0.128$ |

**Table 6 results for crack growth in HAZ and in Air environment**

Figure 20 shows assessment points from initial crack propagation at start of service life to the final year of service. If the service continues beyond the design life (20 years), the structure is likely to fail in elasto-plastic mode, providing reasonable level of plasticity from safety point of view.

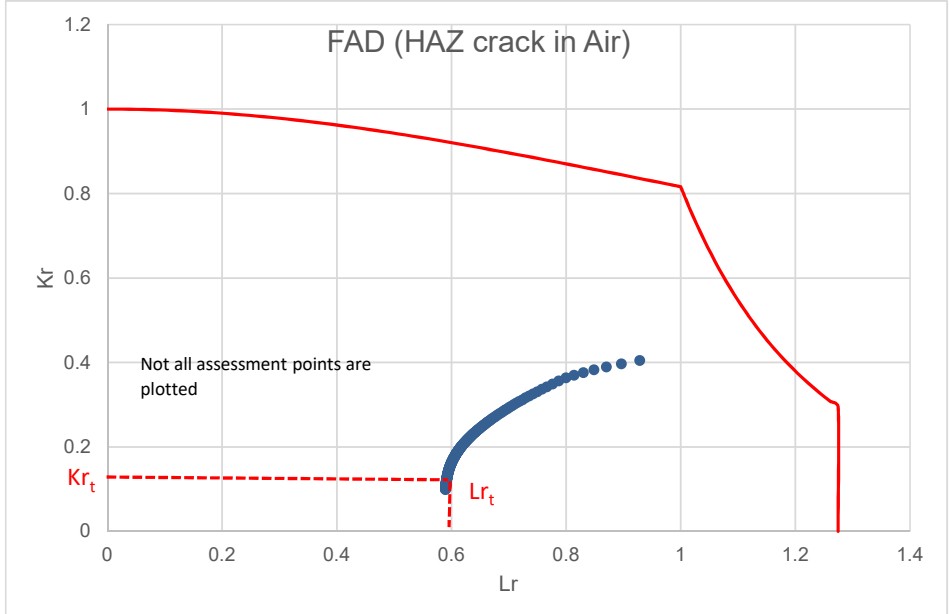

**Figure 20 Failure assessment diagram (FAD) for crack growth in HAZ and in Air environment without inspection**

As explained earlier a damaged tolerant design is closely tied to in-service inspection. Here, it is assumed that a MPI inspection is carried out at year 12. When no crack is detected or repaired if detected, the predicted crack size is updated and reduced back to the initial crack size. This is shown with solid lines after year 12 in Fig. 14. The final year crack size remains below the tolerable limits.

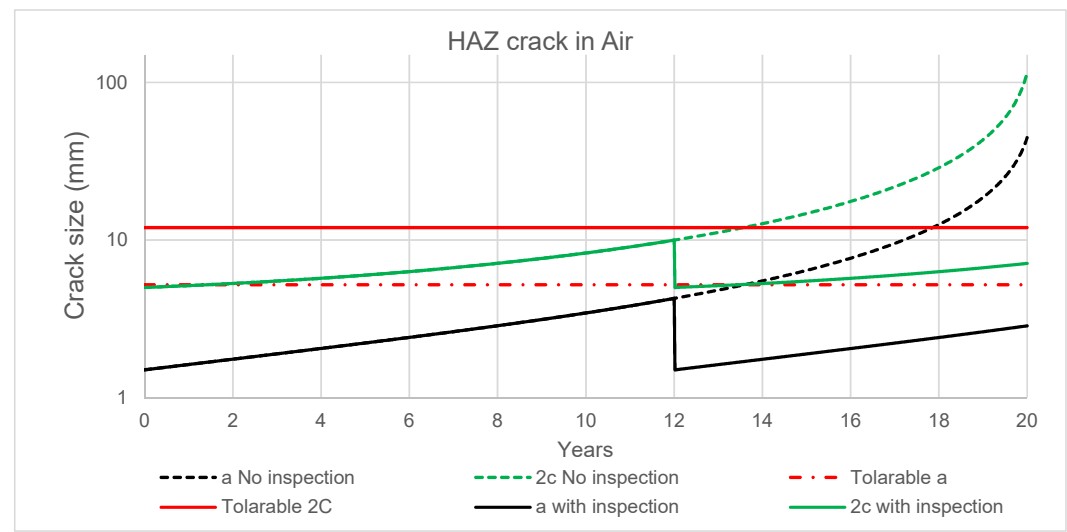


**Figure 21 Crack growth curves for propagation in HAZ and in Air environment**

The weld profile condition may be as‑ welded or ground flushed depending on fabrication
specification and could be altered by the design engineer. The effect of such condition was
studied by considering the influence of weld profile on POD for the MPI method. MPI can find
smaller cracks in the welds with ground flushed crowns (Table 1). As shown in Fig. 21 improving
the weld joint design by specifying ground flushing requirement reduces the inspection
frequency from twice to once in 20 years of service.

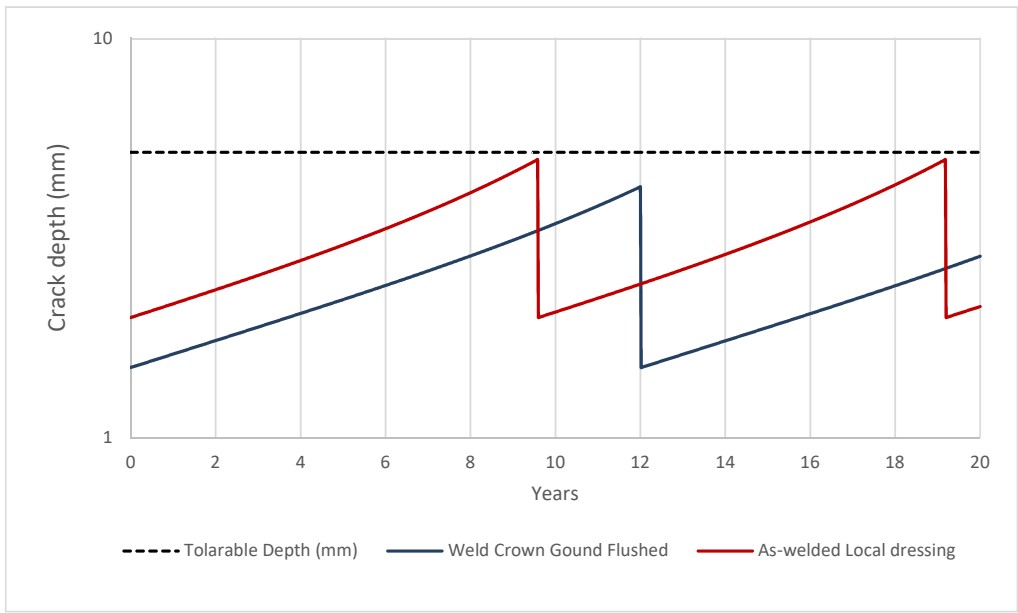


**Figure 22 Effect of weld profile condition on in-service inspection**






The effect of choice NDT for in-service inspection was studied by considering a case were UT is
chosen as the inspection method. The detection reliability specified in Table 1 used to determine
the crack size that can be left undetected after inspection. Figure 22 shows the predicted crack
size compared to inspection with MPI. It is observed that in order to keep the crack size below
tolerable size three inspections are required instead of one inspection using MPI.

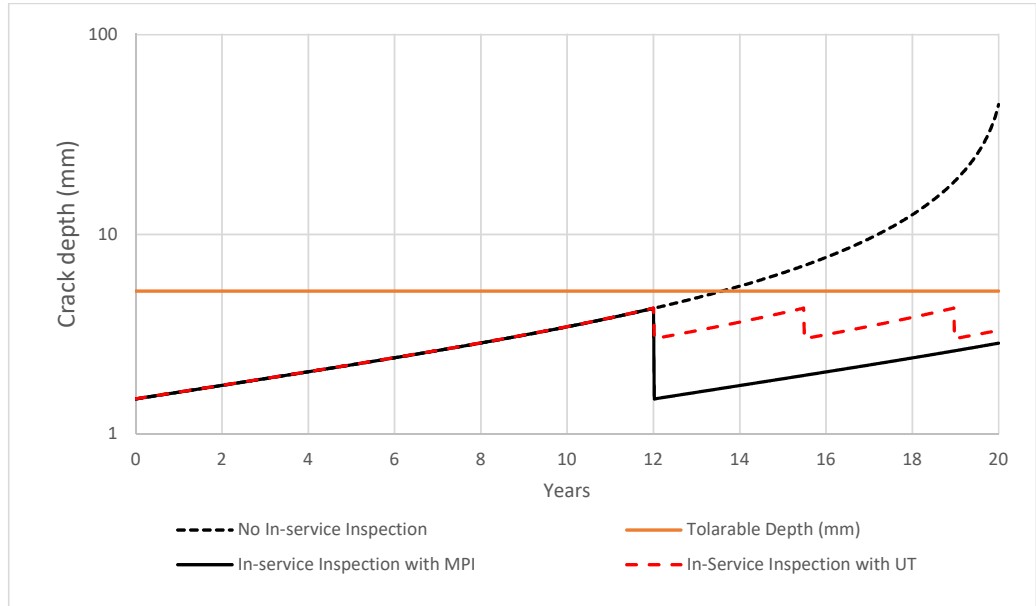


**Figure 23 Selection of NDT method based on probability of detection and crack size at the time of inspection**
## 5.2  Effect of environment
In the event of insufficient corrosion protection, the fatigue crack growth will be accelerated.
The accelerated crack growth rate is reflected in fracture mechanics through changing the Paris
law constants to those observed in corrosive environment. This is shown in Fig. 15 and Fig. 16,
where the previously studied defect is assessed under free corrosion environment instead of the
air environment. It is observed that failure is predicted to occur as early as 3.4 years after
commissioning. One strategy could be an increased attention to execution of corrosion protection
measures prior to commissioning. Additionally the joint should be inspected for the signs of
corrosion at least every three years.

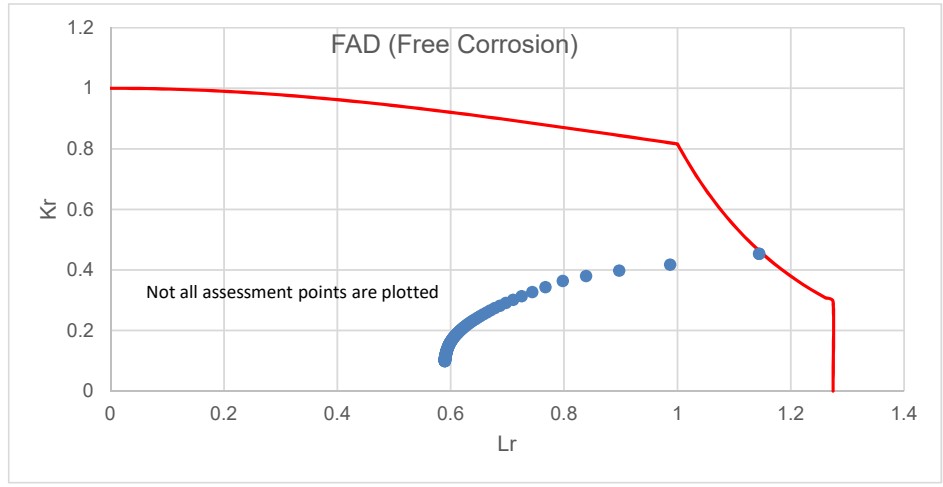

**Figure 24 Failure assessment diagram (FAD) for crack growth in HAZ and with free corrosion**

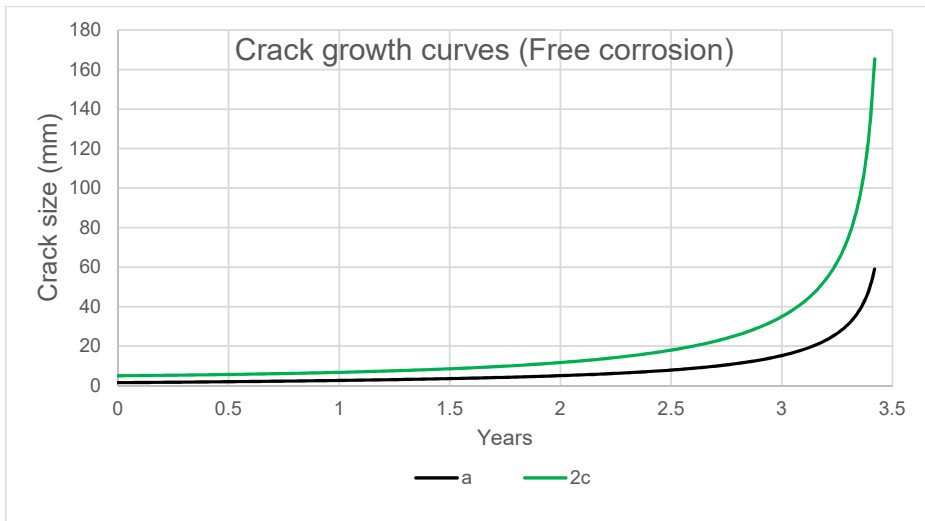


**Figure 25 Crack growth curves for propagation in HAZ and with free corrosion**

## 6   Case-Study 2: Probabilistic Fracture Mechanics application to a plate failure

Many structure members in offshore can tolerate cracks even after they become through
thickness. These structures may be idealised by plates containing through thickness cracks (Fig.
20). This can be for example for a less critical location of the structure in case-study 1 with lower
stress levels.
Here, application of probabilistic fracture mechanics to such a structure is demonstrated. The
assumed inputs are listed in Table 7.



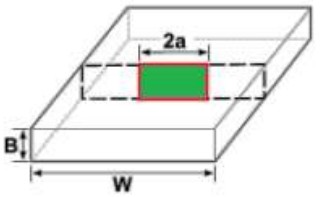
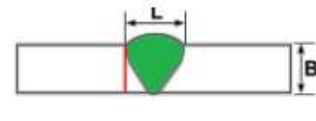

**Figure 26 Through-thickness Crack geometry diagram**

| Case Description | | |
|---|---|---|
| Case study structure | Offshore topside Platform with Long-term stress shape parameter = 0.85 and load cycle rate = 5.063 cycles/ min | |
| | Maximum design stress = 0.62 * Yield stress | |
| Material Properties | Young Modulus | 210 constant |
| | Poisson Ratio | 0.3 constant |
| | Yield stress ($Y_S$) | 450 constant |
| | Tensile strength | 560 constant |
| | Toughness | 200 MPa* m^0.5 assumed |
| Fatigue assumptions | Crack growth model | Single slope Crack growth |
| | Cyclic stress | Equivalent constant amplitude stress 21 MPa |
| | Stress Intensity Solution | Through-thickness flaw in an infinite Plate |
| | Paris Law parameters | BS 7910 recommended values |
| | Design cycles in life | $N_{life} = load\ cycle\ rate\ (\frac{cycles}{min}) * (20\ [year] *$ $365[day\ per\ year] * [hour\ per\ year] *$ $60\ [min\ per\ hour])$, for this structure $= 5.322 * 10^7$ |
| Fracture assumptions | FAD | BS 7910 Option 1 |
| | Primary stress | Weibull distribution with scale parameter 9.47 MPa |
| | Secondary stress | Weld Residual stress= Constant 100 MPa, assumed |
| | Thickness (B) | 60 (mm) |
| | Initial Flaw dimensions (2a ) | Exponential distribution with mean value of 2 mm |
| Inspection Capabilities | In-service surface inspection | Surface inspection for ground welds above water surface (Fig. 10) |

**Table 8 Inputs for probabilistic Fatigue and fracture mechanics assessment**
Figure 21 shows fatigue and fracture reliability of the structure under three levels of equivalent
constant amplitude cyclic stress. As a starting point, 21 MPa cyclic stress which corresponds to
extreme stress of 0.62 $Y_S$ is selected. Target reliability level of 1.00 x 10⁻⁴ from Table 6 for
Offshore Wind Turbines (unmanned structures) is selected. The structure will reach to the
target tolerable probability of failure just before year 17, suggesting that the structure should
be inspected prior this time. As it is shown in Fig. 25, such an inspection will reduce the failure
probability below the target level for the rest of the intended service life.
If the aim was to design the structure to the safe-life design philosophy, the stress would have
needed to be reduced below current level. This, however, may not be an economical option since
the current extreme stress level already possesses significant safety factor (0.62 $Y_S$) and



reducing the stress will require bigger cross sectional dimensions and, hence, a heavier and
more expensive structure. Integrating in-service inspection options in design can potentially
result in a more efficient design.
Furthermore, the design cyclic stress may be increased considering the availability of in-service
inspection. Two stress levels are considered here: An upper bound limit value of 35 MPa
corresponding to extreme stress equal to the Yield stress and a moderate value of 26 MPa. As
depicted in Fig. 21, the probability of failure curve will be shifted to left 2 and 3 years,
respectively. It is evident that the structure can sustain higher levels of stresses provided that
appropriate time for inspection is determined and also other required limit states are not
violated.

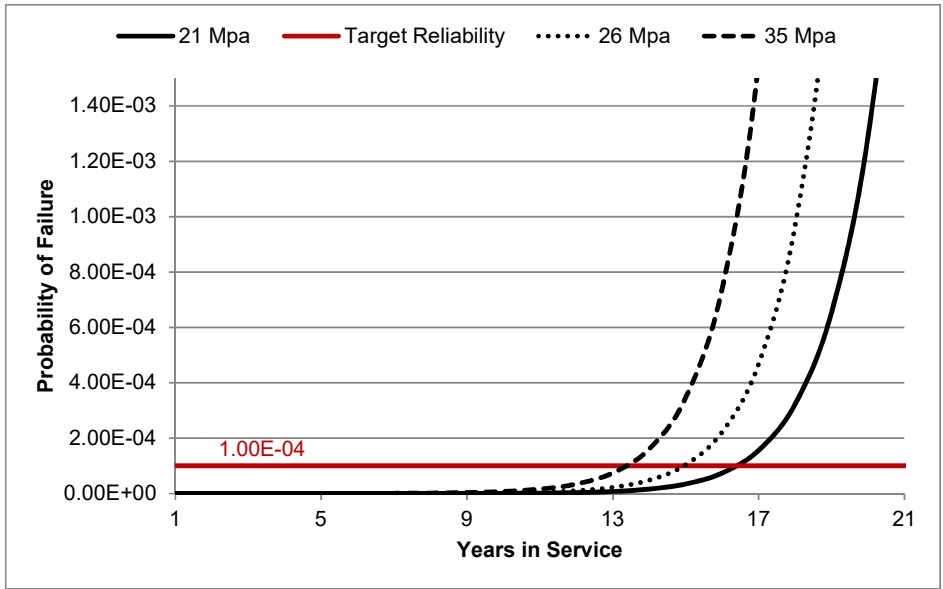

**Figure 27 Fatigue reliability (FM) of a welded joint in an offshore structure for three different constant amplitude**
**stresses**
The effect of an inspection schedule is considered for the case of through-thickness crack under
21 MPa cyclic stress. It was shown previously in Fig. 21 that, the structure is predicted to reach
the target tolerable probability of failure just before year 17, thus, the inspection should be
scheduled prior to this time. Here, a number of inspection options are considered.
Any inspection earlier than year 6 appears to have little benefit as the failure probabilities are
below 5.0E-8, a very low probability of failure. The reduction in probability of failure is in the
order of one and the structure is likely to exceed the target level of reliability again close to the
final year of service. Inspection between year 10 to 15 show the most effective results by keeping
the structure way below the target level throughout and to the end of service life ensuring
considerable level of safety as well as providing further life extension possibilities in the final
years of designed service life.





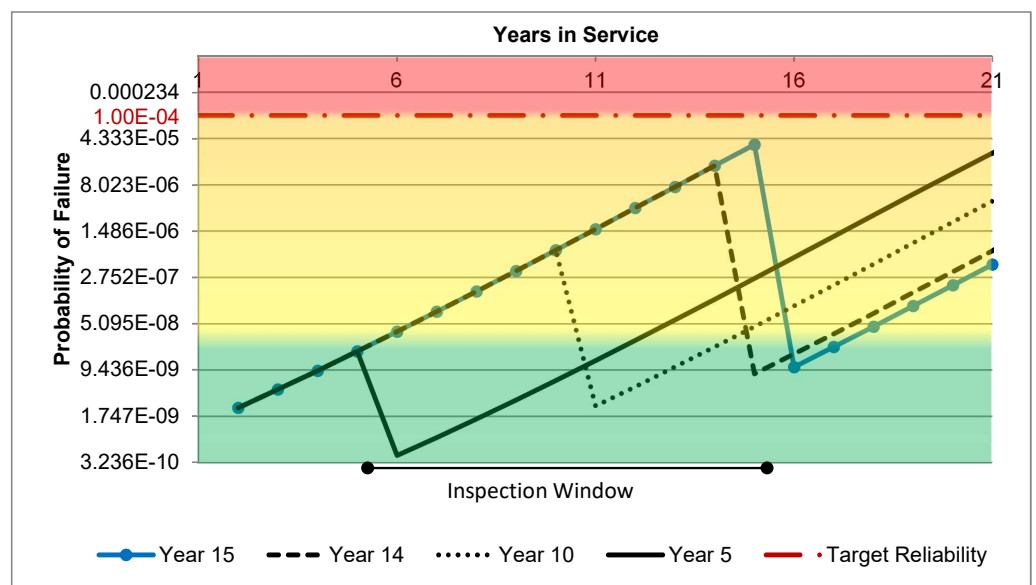

**Figure 28 Crack growth curves of case study through thickness in a plate considering different first inspection times**

# 7    Conclusions

This paper presented a new approach in fatigue design of offshore wind turbine support structures. Traditionally, design of offshore renewable structures against fatigue failure has been performed using the so-called S-N curve method. This approach, however, suffers from a number of limitations, such as limited ability to integrate the inspection capabilities. The structural design can significantly benefit from inspectability of the structure by considering the damage-tolerant nature of many offshore structures. Fracture mechanics is a powerful tool capable of address a wide range limitations associated with of the S-N approach.

In this work, a framework for design of offshore structures based on fracture mechanics was developed and its applications to a monopile wind turbine support structure were demonstrated. Additionally, probabilistic fracture mechanics approach and its application in optimising in-service NDT inspection for a plated structure under see wave loading was presented.

It was found that the design of the structure can be enhanced through specifying weld crown improvements which leads to better fatigue performance and reduced in-service inspection. The Magnetic Particle Inspection (MPI) will require three times less inspection interval than Ultrasonic Testing (UT).

The probabilistic model showed to have the capability to account for uncertainty in design and inspection variables including NDT reliability. It also provides a likelihood of failure which can be used to calculate the risk associated with the chosen inspection time and in turn for optimising inspection using a, for example, cost benefit analysis.

Additionally, the proposed optimisation model can be used for any practice of structural optimisation of OWT support structures





## Authors contribution

PA conducted the research, created the proposed framework, performed all case study analysis, made the figures, and planned and wrote the paper. BF and AK contributed to the research with intensive discussions and added to the paper with conceptual discussions and internal review. AK secured the funding for this paper.

## Competing of interest

The authors declare that they have no conflict of interest.

## Acknowledgments

This work was supported by a grant from the Supergen Wind Hub EP/L014106/1, from the UK Engineering and Physical Sciences Research Council (EPSRC), under the Flexible Funding Scheme for University Strathclyde. Furthermore, this project has received funding from the European Union's Horizon 2020 research and innovation program under grant agreement No. 745625 (ROMEO, 2019).

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
