# Peer review of "A fracture mechanics framework for optimising design and # inspection of offshore Wind Turbine support structures against"

_Wind Energy Science, 2020_

## Referee Comment (RC1) · Arno van Wingerde (Referee) · 17 Jul 2020

There is not that much actual research, either numerical or experimental, presented in the paper, but it serves as an introduction and application. What makes the paper worthwhile is a nice combination of design standards, fracture mechanics and probabilistic approach presented with a practical application.

The first 20 pages paper provides a lengthy albeit still readable introduction on Fracture mechanics and probabilistic design. Consider condensing a bit: reference to a national standard is of limited interest for people using other standards and should just be used to outline the real topic of the paper.

The principle of using fracture mechanics on based probabilistic design is demonstrated applied to a weld in a monopile and a weld in a plate, showing the possibility of the technique. Given the length of the paper a list of symbols and definitions may be useful.

A number of small recommended edits: 19: In-service => in-service 22: showed to possess => showed 26: framework => the framework. In general articles and plurals seem to be missing here and there, the English is Ok but would benefit from a read-over by a native speaker. 37: Turbine (OWT)=> Turbines (OWTs) 38: Social acceptance is also an important driver of OWT, you might wish to mention it here 42: structure => structures 53: approach => approach, 58: is => are 76: POND => PoND suggestion 103: relived => relieved 151: natriting "treat (a substance) with nitric acid, especially so as to introduce nitro groups". I am pretty sure you didn't mean that the an equation. . .. 171: or air-filled offshore structure where the pressure or absence of water inside the structure can be used as a simple way to detect through-thickness cracks. 202: Failure => the Failure 219: Fig. 6 => Fig. 7 Check figure references as this happens several times in the paper. 224: a picture with the ïAş-ïAě diagram of mild steel helps explain this. 228: Fig. 6 => Fig. 7 231-255: the explanation of the various options on the BS which only matters for Lr>0.9 could be omitted altogether 257: figure 7 shows on the axes: Kr against Lr which are material properties: OK for the FALD curve, but not correct for the example assessment points "safe"/"unsafe" 287: size has => size ac has 292: Fig.8 => Fig. 9 295: PODs => PoDs 346: POD => PoD please check the paper for that 471: units missing Young Modulus 210 => Young's modulus 210.000 MPa 480: Primary => The primary 568: posseses => possesses a 568: (0.62 Ys) => (Ys = 0.62) 609: require three times less inspection interval => allow for thrice the inspection window
* * *

---

## Referee Comment (RC2) · Anonymous Referee #2 · 27 Dec 2020

The manuscript presents a fracture mechanics strategy for defining the inspection intervals regarding fatigue crack growth in the supporting steel structures in offshore wind turbines (OWT). Even though the subject is highly relevant for the journal, some improvement needs to be made before it can be recommended for publication. The aim of many of the improvement is aimed for improving the readability of the manuscript.

1) The manuscript contains many abbreviations, and the use of them need to be somehow structured: a. Avoid using abbreviations in the manuscript b. Avoid using abbreviations for terms only used very limited in the manuscript c. Include a list of abbreviations and be careful to define the abbreviations the first time they are used and possible also

repeat them if it is a long time since they have been used last time

2) The fracture mechanical approach is presented as an alternative to fatigue damage accumulations approach, which in the manuscript is denoted an S-N method. Of the reviewer's opinion, they may more be considered as two complementary approaches. The fracture mechanical approach is well suited for describing fatigue crack growth, while the Fatigue damage accumulation approach is more suited for scattered fatigue damage evolution.

3) In line 107, it is stated that the compressive residual stress and shakedown phenomena can be addressed using fracture mechanics. They can also be addressed using the fatigue damage accumulation approach.

4) In line 170, it is stated that the criterion is particularly common for pipelines, pressure vessels, etc. But no-one of those cases is particularly relevant for OWT.

5) There is an inconsistency between equations 9, 10, and line 240: is $Sig\_UTS=Sig\_U$, I $L\_r,max=L\_r$.

6) Equations 11 and 12 can be written much shorter using a split equation with a curly bracket.

7) Equation 15, no definition of J'

8) There are issues with a number of figure and table references: E.g., line 269, 380, 383, 468, 497, 538, 552, 555, . . . ?

9) Figure 9, it is unclear what is defining the length of $l\_1$ and $l\_2$, maybe add the safety factor on the predicted flaw growth curve. They should in someway be correlated with the tolerable flaw size.

10) In line 295, POD is not defined at this point.

11) Figure 10, the second axis in the first and third figure must be a frequency of occurrence and not a probability as it is the actual defect distribution. In addition, the

equation shown in figure 10, must be used the other way as it is the first figure, which contain the unknown.

12) Unclear correlation between figure 11 and table 1 and how is the largest flaw missed by the NDE method (line 327) defined as figure 11 is a continuous function only approaching 1.

13) Is the values are given in table 5 and 8 taken from the Gentils, 2017 reference or where are they coming from.

14) There are missing units on some of the values given in Tables 5 and 8.

15) Maybe the sub-section headline "Crack growth in Air" should be reformulated as a crack can not grow in air.

16) The colors of the lines in figure 21 should be made so it is clear what curves are related.

---

## Editor Comment (EC1) · Lars Pilgaard Mikkelsen (Editor) · 27 Dec 2020

Dear Peyman and co-authors, I apologize for the lengthy review process. Nevertheless, there is now the required two reviews of your manuscript and they are quite positive both of them. Please go through the suggested point for corrections and submit a revised manuscript. Best regards, Lars
* * *

---

## Author Comment (AC1) · 23 Jan 2021

Dear Referee, Thank you very much, indeed, for your constructive comments. My responses to your comments are listed below:

1)Thank you and I agree with I will revise the manuscript and will add a list of abbreviations.

2)I fully agree that the two methods are complementary methods in design and assessment of structures with their own limitations and merits well documented in the literature and also summarised here. The main motivation for a FM based design ap-

proach, here, is the shift of paradigm towards damage-tolerant design philosophy which is best done by considering inspection and maintenance activities. To this end and to optimise design and maintenance information about fatigue and crack size is essential. This information cannot be predicated accurately by only using the S-N method. 3)I will correct this. Thank you!

4)It's "commonly" adopted for application in structures containing containments because of the possible sever consequences (leak or rapture), but can be used for monopiles as well. There is a debate if the structure is capable of sustaining a through thickness crack after full penetration of crack height or the through height failure governs the overall failure. In case study section this was tried to be studied.

5)Equation 9 is Lr and line 240 refers to LrMax . For equation 10, Thank you well spotted! I will change line 240 to $\sigma$U.

6)Will address this in the proof.

7)J is J-integral as also mentioned in line 250 right after equation 16. But, perhaps, it not clear. I will make it clearer.

8)Thank you. I will address these.

9)This length are proposed commonly adopted length the value is recommended to be considerably lower that predicted failure length. And you pointed out can be seen as a safety factor. (line 287-292).

10)Agree. I will remove PODs

11)I believe the figure is correct. If you take the firs figure as the probability- by considering small intervals, the third figure is probability of detected sizes as well. The equation is essentially: P(A I B)= P(A).P(B), where, P(A I B) is probability of finding crack sizes, P(A) is probability of crack sizes being present, and P(B) probability of detecting the sizes.

[Figure]

12)Table 1 is values of POD gives by BS7910 and figure 11 POD proposed by DNV. BS7910 gives largest flaws that can be reliably detected (Typically, 90% probability of detection with 95% confidence). But DNV provides an equation, instead. The largest missed flaw needs to be judged by the assessor- Typically, 80%-95% POD is chosen considering the consequence of possible failure.

13)Yes, those refer to material properties are. I will add the reference to those that are not given.

14)Thank you. I will add the units.

15)This is the common term used BS7910. But you are right, strictly speaking "crack growth is Air environment" is a more accurate term. I will address this.

16)I think that's exactly what they do. Black= a, Green 2C, Red= Tolerable, for Black and Green solid is with inspection and dotted without inspection.

Thank you again, Best regards Peyman Amirafshari

Please also note the supplement to this comment:
https://wes.copernicus.org/preprints/wes-2020-65/wes-2020-65-AC1-supplement.pdf

---

## Author Comment (AC2) · 23 Jan 2021

Dear Dr Arno van Wingerde,

Thank you very much, indeed, for your very insightful and constructive comments. I agree with most of the suggestions and will implement them accordingly. I only have two additional explanations to two of your comments as listed below:

1)Consider condensing a bit: reference to a national standard is of limited interest for people using other standards and should just be used to outline the real topic of the paper.

BS7910 is the only fracture mechanics standard applicable to all metallic structures. E.g. API579 /ASME is for pressurized equipment used in oil & gas, petrochemical, and chemical facilities and R6 is for nuclear facilities. I agree with you that in a scientific paper referencing to standards should be limited but, given uniqueness of BS7910 and the large body of conducted research which has led to creation of the standard and the well-documented references provided in the standard, It is my humble opinion that the reader will benefit from references to the standard. I myself have been a committee member of BS7910 since 2018 and have an internal insight about the extensive research carried out by research bodies particularly TWI.Ltd, that is built into the standard. DNV is also a widely used standard in assessment of offshore structures and perhaps the only standard that provides recommended continuous PoD functions.

2)Kr against Lr which are material properties: OK for the FALD curve, but not correct for the example assessment points "safe"/"unsafe" .

Kr is the ratio of applied crack driving force to fracture toughness, neither are material properties only even Kmat is affected by material thickness therefore a property of the structure. L r is the ratio of applied stress to yield strength and therefore not just a material property.

Best regards, Peyman Amirafshari

Please also note the supplement to this comment:
https://wes.copernicus.org/preprints/wes-2020-65/wes-2020-65-AC2-supplement.pdf
* * *

---

## Author Comment (AC3) · 23 Jan 2021

Dear Lars, Thank you very much, indeed, for considering this paper and your efforts. I have reviewed the comments from the reviewers and as you mentioned they have both made very constructive comments and I agree with great majority of the suggestions. I have posted explanations to some the comments made by the reviewers. I will implement the corrections and submit an updated version. Many thanks, Best regards, Peyman

---

## Editor Comment (EC2) · Lars Pilgaard Mikkelsen (Editor) · 13 Feb 2021

Thank you for your response to the reviewer comments. I'm looking forward to your revised manuscript, preferable in two versions: One with marked corrections and one clean version for publication.

Best regards, Lars